# Antibacterial and Physical Properties of PVM/MA Copolymer- Incorporated Polymethyl Methacrylate as a Novel Antimicrobial Acrylic Resin Material

**DOI:** 10.3390/molecules27248848

**Published:** 2022-12-13

**Authors:** Christopher Lai, Ashten Nguyen, Lynna Ye, Jessica Hao, Hyun Koo, Francis Mante, Fusun Ozer

**Affiliations:** 1Penn Dental Medicine, University of Pennsylvania, Philadelphia, PA 19104, USA; 2College of Arts and Sciences, University of Pennsylvania, Philadelphia, PA 19104, USA; 3Department of Orthodontics and Divisions of Pediatric Dentistry and Community Oral Health, Penn Dental Medicine, University of Pennsylvania, Philadelphia, PA 19104, USA; 4Department of Restorative Dentistry, Penn Dental Medicine, University of Pennsylvania, Philadelphia, PA 19104, USA

**Keywords:** polymethyl methacrylate, acrylic resins, Gantrez, physical properties, antibacterial properties, surface hardness, flexural strength, water sorption, water solubility, orthodontic appliances

## Abstract

Polymethyl methacrylate (PMMA), an acrylic resin used in orthodontic appliances and removable dentures for its biocompatibility and esthetics, may harbor bacteria on its surface. The present study investigated a new PMMA formula with Gantrez: an antibacterial copolymer of methyl vinyl ether and maleic acid (PVM/MA). Samples were tested for mechanical properties (surface hardness, flexural strength, water sorption, and water solubility) and effects against *Streptococcus mutans*. Six groups (0%-control, 5%, 10%, 15%, 20%, and 25% Gantrez) of *n* = 12 were fabricated for physical property tests and analyzed with one-way ANOVA and Prism 6. From these results, three groups (0%, 5%, and 10% Gantrez) were selected for antibacterial tests, and data were analyzed with one-way ANOVA and Tukey’s multiple comparison test. Adding 5% and 10% Gantrez into PMMA significantly decreased *S. mutans* adhesion. There was no significant difference between the control vs. 5%, 10%, 15%, and 20% Gantrez (*p* > 0.05) for surface hardness, the control vs. 5% Gantrez (*p* > 0.05) for flexural strength, and the control vs. 5 and 10% Gantrez for water sorption and water solubility. Overall, incorporating 5% Gantrez into PMMA may be a promising solution to reduce bacterial adhesion without changing the acrylic resin’s physical properties.

## 1. Introduction

The increase in patients seeking orthodontic treatment in recent years has created an unparalleled demand for the use of acrylic resins in orthodontics as well as prosthodontic appliances [1]. The most common acrylic resin in clinical dentistry is polymethyl methacrylate (PMMA), which serves a major role in making orthodontic retainers and dentures in dental laboratories, relining dentures, and temporary crowns in dental clinics, and fabricating artificial teeth in the industry [2,3,4,5]. This material is conventionally available as two components: one in the form of a clear polymer (PMMA, with color additives and an initiator) and the other in the form of a liquid (methyl methacrylate monomer with cross-linking agents and inhibitors). The powder and liquid are mixed to start the chemical reaction that is completed by hardening either through chemicals (self-cure) or through the application of heat (heat cure) [4,5,6].

Due to the considerable number of microorganisms in the mouth, the use of PMMA exacerbates the potential for plaque and microorganism retention by making oral hygiene more difficult to maintain [7]. Specifically, the porous and irregular surfaces of dentures and orthodontic appliances fabricated from PMMA promote biofilm formation and may contribute to dental caries, gingival inflammation, and periodontal disease [8,9]. Geriatric patients or those with poor health are more susceptible to these diseases because they suffer cognitive impairment, reduced motor dexterity, and memory loss [10]. In addition to the potential negative effects on the patient, the retention of bacteria on PMMA may lead to the gradual deterioration of the material with long-term use [2,11]. To limit potential dental diseases and extend the longevity of PMMA, there should be a greater focus on novel methods to inhibit biofilm formation on these acrylic surfaces.

Several traditional strategies are used to limit the bacterial load that develops on dentures and orthodontic appliances. For removable appliances with acrylic materials, such as Hawley retainers, there are a number of methods for hygiene and disinfection. The methods include mechanical brushing with toothpaste or soap, soaking in disinfectant cleansing solutions, and physical-chemical methods using an ultrasound. A comparative study showed that a combination of both mechanical and chemical cleansing methods results in more effective biofilm removal. However, such measures rely generally on the conformity of patients and may not be optimal for the treatment of pediatric, geriatric, and handicapped individuals [12,13]. In addition, for fixed orthodontic appliances, such as the Haas, Nance, thumb, or tongue thrusting habit appliances, the methods are even more limited since they cannot be removed from the patient’s mouth.

Recently, several emerging novel strategies have been proposed to reduce the risk of dental caries during dental treatment. However, only a few focus on *Streptococcus mutans*, which is the primary etiological group of biofilm-mediated dental caries [14]. Of these studies, an even fewer number can be applied to fixed orthodontic appliances. Some examples of possible studies include the addition of propolis nanoparticles [15], TiO_2_ and CuO nanoparticles [16], and doxycycline [17]. An additional additive that inhibits *S. mutans* deposits while maintaining its biocompatibility remains highly desirable [18].

There is active research on a copolymer of methyl vinyl ether (PVM) and maleic acid (MA), also commercially known as Gantrez, which reduces plaque retention by preventing initial bacterial adhesion to enamel surfaces through electrostatic repulsion and the release of calcium ions [19]. In the polymerization process, an anhydride form (Gantrez-AN) is formed from a free carboxylic acid monomer. In addition to preventing plaque formation, the PVM/MA copolymer (Gantrez), when incorporated into a mouthwash, reduces in vitro biofilm formation by 31% [20]. This antibacterial additive may prove to be an effective addition to the PMMA material that is used for orthodontic appliances and dental prostheses by reducing bacterial load and lowering plaque and biofilm formation. Due to the aforementioned benefits, the present in vitro experiment explores the physical and antibacterial effects of combining Gantrez with PMMA. Gantrez was expected to enhance the antibacterial properties of PMMA while minimally impacting its mechanical properties, and the null hypothesis was that Gantrez would not have an antibacterial effect on PMMA.

## 2. Results

### 2.1. Surface Hardness (Vickers) Test

Figure 1 and Table 1 show the surface hardness values calculated for the sample groups (0%, 5%, 10%, 15%, 20%, 25% Gantrez) comparing the hardness values at 30 min after fabrication and after 24 h of submersion in 37°C dH_2_O. Compared with the control group, a statistically significant difference (*p* < 0.05) was observed for only Group F (25% Gantrez). The highest hardness value (11.67) was for group B (5% Gantrez), and the lowest (8.07) was for group F (25% Gantrez).

### 2.2. Flexural Test

Figure 2 and Table 2 show the flexural strength (MPa) calculated for the sample groups (0%, 5%, 10%, 15%, 20%, and 25%). Compared with the control group, a statistically significant difference (*p* < 0.05) was observed for Groups C to F (10%, 15%, 20%, and 25% Gantrez). There is no significant difference for Group B (5% Gantrez). The highest flexural strength value (66.19 Mpa) was for the control group (0% Gantrez), with decreasing values at the lowest (41.45 MPa) for group F (25% Gantrez).

### 2.3. Water Sorption and Water Solubility Tests

Figure 3 and Table 3 show the water sorption and water solubility calculated for the sample groups (0%, 5%, 10%, 15%, 20%, and 25%). Compared with the control group, a statistically significant difference (*p* < 0.05) was observed for Groups D to F (15%, 20%, and 25% Gantrez). There is no significant difference between Group B and C (5% and 10% Gantrez). Group F (25% Gantrez) showed the highest water sorption (160.92 μg/mm^3^) and solubility (188.51 μg/mm^3^). As the concentration of Gantrez decreased, the experimental values decreased to the lowest water sorption (17.40 μg/mm^3^) and water solubility (10.60 μg/mm^3^) values for group A (control).

### 2.4. Microbiology Test

The bacterial viability (CFU mg-1) for the three different concentration groups- A (control), B (5% Gantrez), and C (10% Gantrez)- were tested at dilutions of 1 × 10^5^ and 1 × 10^6^. Groups D (15% Gantrez) and E (20% Gantrez) were not tested in this study because of the impaired flexural strength properties demonstrated by the prior physical tests. In addition, Group F (25% Gantrez) was not tested because it showed both impaired flexural strength and Vickers hardness. Dry weight values (g) were calculated for the three groups (Control, 5% Gantrez, 10% Gantrez), and the control group had the highest dry mass value (Table 4). Overall, the microbiology tests revealed that there was a significant decrease in the number of colonies as the concentration of Gantrez increased (Figure 4).

## 3. Discussion

### 3.1. Vickers Hardness

Surface hardness is a commonly studied physical property for determining the durability of dental materials inside the oral cavity [21]. It is the ability of the material to resist the plastic deformation caused by penetration, indentation, scratching, and abrasion [22]. In the oral cavity, materials are exposed to proteins, polysaccharides, microorganisms, and food substances, leading to complex mechanical interactions that may impair the longevity of the dental material. The Vickers hardness test measures the resistance of acrylic resin materials when a load is applied over the surface area of indentation, thus providing a reliable assessment of the surface hardness [11,23].

In the present study, measurements were taken on well-polished samples immediately after the indentation was made since an accurate measurement greatly depended on a material’s elastic recovery and surface homogeneity. The surface hardness of the prepared acrylic samples showed that increasing Gantrez concentrations up to 20% did not significantly affect the wear resistance compared to the control. In some instances, Gantrez even reinforced PMMA, yielding higher hardness values after 24 h for groups B (5% Gantrez) and C (10% Gantrez) compared to the control. However, further increasing of the Gantrez concentration to 25% decreased the hardness value after 24 h from 11.09 (control) to 8.0, which was a significant difference as calculated by one-way ANOVA and Tukey multiple comparison tests. This suggested that when higher Gantrez concentrations were incorporated, the surface of PMMA was more susceptible to becoming rough, which resulted in plaque accumulation and microporosities. Furthermore, the material will be at a higher risk of discoloring and fracturing due to pigments entering the porosities [24]. Therefore, based on the surface hardness results, 5–20% Gantrez may be added to PMMA without negatively impacting the material’s ability to withstand the brushing, chewing, and cleaning that often occur in the oral cavity.

### 3.2. Flexural Strength

Despite acrylic PMMA’s popularity in dentistry, the material can easily fracture due to its low resistance to impact, low flexural strength, and low fatigue strength [25,26]. Acrylic resin-based orthodontic appliances and removable prostheses, such as dentures, can often fracture from repeated masticatory loads and the frequent removal and reinsertion of the appliances. Thus, it is important that additives to PMMA do not further decrease the flexural strength of the material.

In the present study, the flexural test data of the control group (66.19 ± 7.48 MPa) were consistent with previously published values, which demonstrate the flexural strength of pure PMMA to be 62.95 ± 7.32 MPa [27]. By incorporating Gantrez, there was no significant difference in the flexural strength between the control and Group B (5% Gantrez). However, when starting to incorporate 10% or more Gantrez into the acrylic resin, the flexural strength started to decrease significantly. At higher concentrations of 20–25% Gantrez, the flexural strength decreased under 50 MPa, which is the standard minimum limit for the flexural strength of acrylic resin as outlined by ISO 20795-1 (2008) [28]. This suggests that by introducing higher concentrations of this new component into PMMA, there is more possibility of fracture in the dental prosthesis or orthodontic appliance.

### 3.3. Water Sorption and Water Solubility

Water sorption, a physical property of acrylic resins, results in dimensional instability due to internal stressors. Oftentimes, these stressors are created by water molecules, which enter and force a material’s macromolecules to separate. This may lead to cracks and future fractures within dental prostheses and orthodontic appliances [4,29,30,31]. Therefore, water sorption is a critical assessment of a material’s durability because it predicts whether the oral environment will have a negative effect on the material’s stability. In addition, the water solubility of dental acrylic resins results from traces of unreacted monomer and water-soluble additives leaching out into the oral fluids. The escaped compounds may react with the surrounding soft tissue, and thus, it is important to determine the solubility of these test materials [32].

The results of the water sorption and water solubility studies for 15% Gantrez and beyond produced significantly different results from the control group. The data suggested that when increased amounts of Gantrez were combined with PMMA, the dental acrylic absorbed more water and became more soluble. This was due to more microgaps or less bond formation within the PMMA material, which was readily observed by the authors during sample fabrication as the surface appearance became more porous when more Gantrez was added. The samples saw a steeper increase in water solubility with 20% and 25% Gantrez, which may be explained by Gantrez interfering with PMMA bond formation at higher concentrations. This phenomenon can be problematic for dental material, especially if it is meant to be used long-term since it will potentially lead to fractures and soft tissue reactions if the material is cytotoxic. However, with the addition of 5% and 10% Gantrez, there was no significant difference in water sorption and solubility from the control group.

### 3.4. Antibacterial Properties

The presence of acrylic-based orthodontic appliances and removable prostheses can create new retentive areas and surfaces that induce the local adhesion and growth of *S. mutans* [27,33]. The adhesion of microorganisms to the acrylic material’s surface initiates the formation of biofilms. To mimic the oral environment, all bacterial biofilms in this study were formed on salivary pellicles established on acrylic surfaces. In the oral cavity, the acquired pellicle is a biological layer produced by the selective adsorption of salivary proteins on tooth surfaces. This layer is necessary for the attachment of oral bacteria, such as the *S. mutans,* which is focused on in the present study [34].

The results indicate that there is a significant decrease in the number of bacterial colonies and biofilm formation with the increasing proportion of Gantrez incorporated in the material. The decrease in the dry weight suggested there is less bacterial biomass that developed on the surfaces of the acrylic materials. The data gathered from the microbiological tests suggest that Gantrez-containing acrylic resins exhibit strong antibacterial properties.

## 4. Materials and Methods

### 4.1. Sample Preparation with Various Ratios of PMMA and Gantrez

Six groups of 10 × 2 mm circular disks (*n* = 12) were prepared based on the manufacturer’s instructions with a previously fabricated plastic mold. Specific amounts of PMMA and Gantrez S-97 (Ashland Global, Wilmington, NC, USA) were weighed using an analytical scale with an accuracy of 0.0001 g (Table 5). A thin layer of petroleum jelly was first applied on the internal surfaces of the plastic mold. The volume of the liquid PMMA monomer was standardized using a graduated cylinder and was held constant throughout the experiment. This liquid monomer was then hand-mixed with the specific weights of PMMA and Gantrez from each group. Prior to adding the dry powders of PMMA and Gantrez to the liquid monomer, the powder was mixed by hand until homogenous. To adhere to the same procedure and maintain consistency throughout the samples, the control group with PMMA powder alone was also hand-mixed. The samples were then placed into a room temperature pressure pot (Aquapres™ Pressure Pot; Lang Dental Manufacturing Co., Wheeling, IL, USA) at 30 psi for 30 min. Finally, the samples were polished with 600, 800, and 1200-micron metallographic grinding paper (LECO Corporation, St. Joseph, MI, USA) until they were flat with smooth surfaces.

### 4.2. Surface Hardness (Vickers) Test

Two sets of measurements were taken for the surface hardness test using a microhardness tester machine (Model type M-400-G1; LECO Corporation, St. Joseph, USA). The first measurement was taken one hour after fabrication and the second after emersion in deionized water for 24 h to compare the change in the surface hardness over one day in a solution. A total of three randomly selected indentations were made for each of the 12 samples per test group. The value of the load (50 gm) and the duration of time (30 s) applied were standardized for each specimen. The lengths of the indentation were measured immediately after they were made to minimize the possibility of viscoelastic recovery. Vickers hardness was automatically calculated and then averaged among the three values. Each data set was further analyzed using one-way ANOVA and Tukey’s multiple comparison test at α = 0.05.

### 4.3. Water Sorption and Water Solubility Tests

Six groups of 10 × 2 mm circular disks (*n* = 12) were used for the water sorption and water solubility tests. Prior to the tests, each sample’s radius and height were accurately measured using a digital caliper to calculate the volume of each sample.
V = πr^2^h(1)
where V = the calculated sample volume (mm^3^); π = estimated as 3.14; r = calculated sample radius (mm); h = measured sample height (mm).

The samples were maintained at room temperature for 30 min after fabrication and were immediately weighed using an analytical scale with an accuracy of 0.0001 g. This weight value was considered the initial weight of the specimen (M_1_). The samples were then placed into a dH_2_O bath and maintained at a constant 37 °C. All specimens were weighed every 24 h until a constant weight (M_2_) was achieved. Prior to weighing, the samples were air-dried completely until no water was seen on their surfaces. After a constant weight was reached, the samples were dried in a vacuum oven in silica gel at 37 °C overnight and weighed daily until a constant weight (M_3_) was achieved. The values for water sorption (Wsp) and solubility (Wsl) in g/mm^3^ were calculated for each sample using the following equations [35].
Wsp = (M_2_ − M_3_)/V(2)
Wsl = (M_1_ − M_3_)/V(3)
where M_1_ = conditioned mass prior to water immersion (g); M_2_ = sample mass after water immersion (g); M_3_ = reconditioned sample mass after drying (g); V = calculated sample volume (mm).

The mean values of the water sorption and water solubility properties were evaluated using one–way ANOVA and Tukey’s multiple comparison test at α = 0.05.

### 4.4. Flexural Test

The flexural modulus of Gantrez-containing orthodontic acrylic specimens was evaluated according to the specifications outlined in ISO 20795-2 [36,37]. Silicone molds were prepared using 64 × 10.0 × 3.3 mm waxed patterns, and acrylic strips were prepared within the silicone molds following the manufacturer’s instructions. A total of 12 specimens free of porosity were selected for flexural testing, and each strip was polished with 600, 800, and 1200-micron metallographic grinding paper until both surfaces were flat and smooth. The resulting width and height of the bars were recorded, and the specimens were stored in deionized water at 37 °C for 50 ± 2 h prior to flexural testing.

The specimens were retrieved from water storage, dried using an air syringe, and placed on two supporting rods 50 mm apart from each other. Using a universal testing machine (Instron no. 4204; Instron Corp, Norwood, MA, USA), a 3-point bending test was conducted on the samples. An increased loading force was applied through a universal testing machine at a constant displacement rate of 5 ± 1 mm/min until the specimens fractured. The load at fracture was used to determine the flexural strength via the equation [38]:(4)$$\sigma =\frac{3\mathrm{Fl}}{2{\mathrm{bh}}^{2}}$$
where F = the maximum load exerted on the specimen (N); l = the distance between the supports (50 mm); b = the specimen width (mm); h = specimen height (mm).

Each data set was further analyzed using one-way ANOVA and Tukey’s multiple comparison test at α = 0.05.

### 4.5. Microbiology Test

Four 2 × 2 cm discs were fabricated with plastic molds using PMMA and each of the two Gantrez concentrations: 5% and 10%. The disc in the molds was held between mylar strips and glass slabs before light curing in order to obtain standard and smoothest resin surfaces. The polymer alone and hydroxyapatite discs were used as controls for the study, with four samples from each group. Altogether, there were 16 discs, which were sterilized after fabrication and transferred into a sterile 24-well tissue culture plate.

Next, a healthy human subject (female, age 27) who had not been under antibiotic therapy for at least a year was asked to chew paraffin film to stimulate salivary flow. A total of 30 mL of freshly stimulated saliva was collected from the subject in the morning, and 2.8 mL of homogenized saliva was dispensed into each of the 16 disc-containing wells. After the tissue culture plate was incubated for 1 h at 37 °C, Streptococcus mutans UA159 biofilms formed on the saliva-coated discs. The discs were then placed in a vertical position in batch cultures at 37 °C and 5% CO_2_. Biofilms of *S. mutans* were formed in the ultra-filtered (Amicon 10 kDa molecular weight cut-off membrane; Millipore Co., Burlington, VT, USA) tryptone-yeast extract broth with 1% sucrose. The medium was changed at 8 am and 6 pm, and the pH of the medium was measured. Afterward, the biofilms were allowed to grow undisturbed for 44 h. Finally, they were analyzed for biomass (dry weight), and the bacterial viability was determined by counting the colony-forming units (CFU). Finally, the data were statistically analyzed with one-way ANOVA and Prism 6 tests.

## 5. Conclusions

The incorporation of Gantrez into acrylic resin results in strong antibacterial activity against S. mutans adhesion to the formed acrylic surfaces. This can potentially reduce biofilm formation and ultimately alleviate dental diseases for those with dental prostheses and orthodontic appliances. Although antibacterial activity was more pronounced as Gantrez concentration increased, the concentration of the additive was limited by its effects on the physical properties of PMMA. For example, the surface hardness decreased significantly when concentrations were greater than 20%. In addition, the flexural strength of PMMA decreased as more Gantrez was added, and the only concentration that had no significant difference compared to the control was 5%. Finally, only between 5 and 10% of Gantrez showed no difference in water sorption and water solubility compared to the control. Based on the following results, the authors recommend adding 5% Gantrez to PMMA in order to enhance the antibacterial effects of dental prostheses and orthodontic appliances without diminishing their physical properties.

Although the preliminary results of the antibacterial tests are promising, more investigations must be conducted before determining the impact that the real conditions of the oral environment will have on the novel material. In other words, due to the complex nature of the oral environment, there are a variety of other factors and conditions that were unable to be mimicked by the in vitro tests performed in this study. For example, there are a host of other types of microorganisms present in the oral cavity in addition to *S. mutans*, and it would be valuable to observe the effect of Gantrez-incorporated PMMA on other types of microorganisms. In addition, since the microbiology experiments in the present study only involve salivary pellicles, it would be beneficial to perform additional experiments with both cultures to support the current findings. Finally, due to the promising results, further in vivo evaluations in orthodontic or prosthodontic patients may ensue.

## Figures and Tables

**Figure 1 molecules-27-08848-f001:**
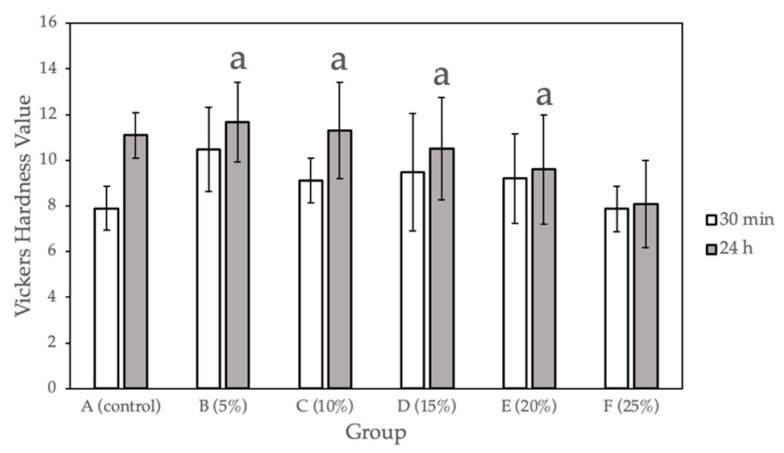
Vickers hardness for self-cure PMMA with various Gantrez concentrations (*n* = 12). Note: means with letter “a” did not have a statistically significant difference (*p* > 0.05) compared to their respective controls.

**Figure 2 molecules-27-08848-f002:**
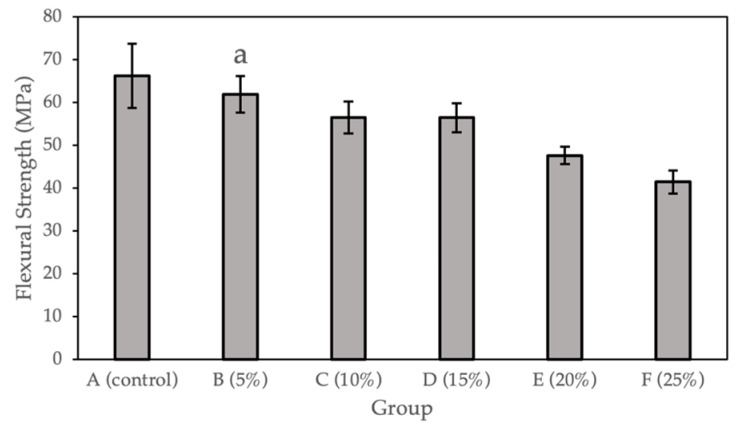
Flexural strength for self-cure PMMA with various Gantrez concentrations (*n* = 12). Note: the letter a represents the absence of a statistically significant difference between the mean of the sample group compared to the flexural strength control (*p* > 0.05).

**Figure 3 molecules-27-08848-f003:**
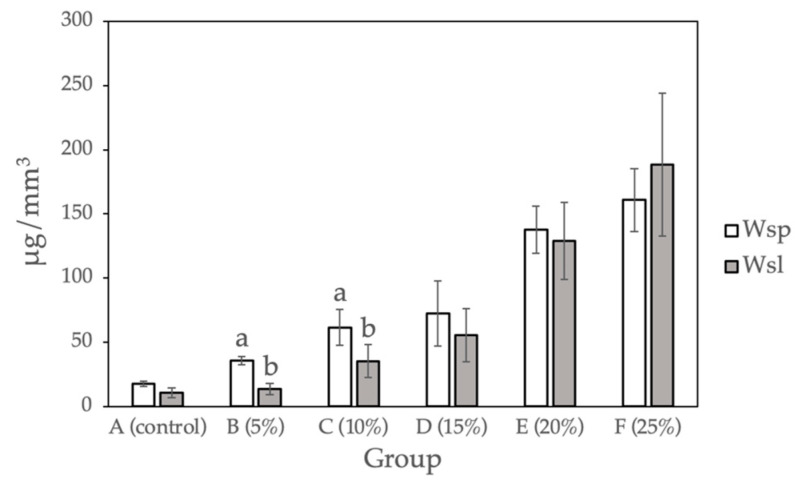
Water sorption and water solubility for self-cure PMMA with various Gantrez concentrations (*n* = 12). Note: the letter a represents the absence of a statistically significant difference between the mean of the sample group compared to the water sorption control (*p* > 0.05). The letter b indicates that there is no statistically significant difference between the mean of the sample groups compared to the water solubility control (*p* > 0.05).

**Figure 4 molecules-27-08848-f004:**
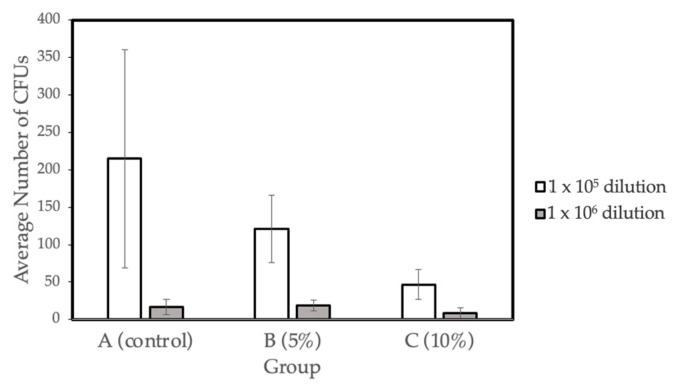
Biofilm formation on self-cure PMMA with various Gantrez concentrations (*n* = 4).

**Table 1 molecules-27-08848-t001:** Vickers hardness for self-cure PMMA with various Gantrez concentrations (*n* = 12).

Group	Vickers Hardness Value at 30 min (SD)	Vickers Hardness Value at 24 h (SD)
A (control)	7.90 (0.97)	11.09 (0.99)
B (5%)	10.47 (1.84)	11.67 (1.74)
C (10%)	9.10 (0.98)	11.30 (2.12)
D (15%)	9.48 (2.56)	10.51 (2.24)
E (20%)	9.20 (1.97)	9.60 (2.39)
F (25%)	7.87 (1.00)	8.07 (1.91)

**Table 2 molecules-27-08848-t002:** Flexural strength for self-cure PMMA with various Gantrez concentrations (*n* = 12).

Group	Flexural Strength (SD)
A (control)	66.19 (7.48)
B (5%)	61.90 (4.26)
C (10%)	56.51 (3.70)
D (15%)	56.48 (3.38)
E (20%)	47.63 (2.04)
F (25%)	41.45 (2.71)

**Table 3 molecules-27-08848-t003:** Water sorption and water solubility for self-cure PMMA with various Gantrez concentrations (*n* = 12).

Group	Water Sorption (Wsp)	Water Solubility (Wsl)
A (control)	17.40 (2.14)	10.60 (3.79)
B (5%)	35.67 (3.41)	13.32 (4.29)
C (10%)	61.37 (14.10)	35.08 (12.86)
D (15%)	72.46 (25.20)	55.31 (20.71)
E (20%)	137.47 (18.29)	129.05 (29.89)
F (25%)	160.92 (24.45)	188.51 (55.63)

**Table 4 molecules-27-08848-t004:** Biomass (dry weight) averages for various concentrations of self-cure Gantrez.

Group	Dry Weight Average (g) (SD)	Significance
A (Control)	1.99 (0.2)	---
B (5%)	1.66 (0.3)	YES
C (10%)	1.77 (0.3)	YES
D (15%)	* Not tested due to significant difference in flexural strength.	
E (20%)	* Not tested due to significant difference in flexural strength.	
F (25%)	* Not tested due to significant difference in flexural strength and Vickers hardness	

**Table 5 molecules-27-08848-t005:** Formulation ratios of PMMA and Gantrez.

Group	PMMA	Gantrez
A (control)	0.500	0.000
B (5%)	0.475	0.025
C (10%)	0.450	0.050
D (15%)	0.425	0.075
E (20%)	0.400	0.100
F (25%)	0.375	0.125

## Data Availability

Not applicable.

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
