# Peer review of "Antibacterial and Physical Properties of PVM/MA Copolymer- Incorporated Polymethyl Methacrylate as a Novel Antimicrobial Acrylic Resin Material"

_molecules, 2022, doi:10.3390/molecules27248848_

Round 1

Reviewer 1 Report

The authors designed and prepared a new PMMA structure, and tested the antibacterial and physical properties of PVM/MA copolymer as a new antibacterial acrylic resin material. The mechanical properties (surface hardness, bending strength, water absorption and water solubility) of the samples and their effects on streptococcus showed that there was no significant difference between the samples and the control group. It proves that the author's work is fruitful.However, there are still some problems to be solved. First, the original intention of the author's research structure needs further explanation; Then, the formation mechanism of this kind of materials is discussed. In particular, the author needs to provide evidence of the prepared materials, such as test data and charts, to enrich the article. As for the antibacterial performance test in this paper, it can be appropriately weakened.

Author Response

Dear Reviewer,

Thank you very much for offering your suggestions for our article titled "Antibacterial and physical properties of PVM/MA copolymer-incorporated polymethyl methacrylate as a novel antimicrobial acrylic resin material". We really appreciate your helpful feedback and valuable suggestions. Below is our response to your suggestions:

Suggestion #1: First, the original intention of the author's research structure needs further explanation; Then, the formation mechanism of this kind of materials is discussed. In particular, the author needs to provide evidence of the prepared materials, such as test data and charts, to enrich the article.

  • Response: Additional charts regarding the data obtained from the tests have been added to the manuscript.

Suggestion #2: As for the antibacterial performance test in this paper, it can be appropriately weakened.

  • Response: A clarification is needed for the comment of the reviewer.

Reviewer 2 Report

The paper deals with antibacterial and physical properties of polymethyl methacrylate with variable addition of PVM/MA copolymer till 20%. It should be better explained why Vickers hardness for all samples after 24 h of submersion in water are superior to the original ones, even for  the control sample, and why the high addition of PVM/MA copolymer diminishes this effect. It should be better explained the clear difference in  Wsp/Wsl behavior between the samples with 20% and respectively 25% addition of PVM/MA copolymer, i.e. what explains the sudden increase of solubility. Maybe some evolution curbs vs.  addition of PVM/MA copolymer in all cases would be more effective in understanding the PVM/MA copolymer influence. As regards the antibacterial properties, the preliminary results seems to be promising, but it is obvious that more investigations are needed to conclude on the real impact under real conditions of oral environment. On the other hand, the authors should detail the processability of the material with e.g. 10% addition of PVM/MA copolymer comparing to usual polymethyl methacrylate. Even if some general details are provided under 'materials preparation', for current practice such details are considered necessary.  

Author Response

Dear Reviewer,

Thank you very much for offering your suggestions for our article titled "Antibacterial and physical properties of PVM/MA copolymer-incorporated polymethyl methacrylate as a novel antimicrobial acrylic resin material". We really appreciate your helpful feedback and valuable suggestions. Below is our response to your suggestions:

Suggestion #1: The paper deals with antibacterial and physical properties of polymethyl methacrylate with variable addition of PVM/MA copolymer till 20%. It should be better explained why Vickers hardness for all samples after 24 h of submersion in water are superior to the original ones, even for the control sample, and why the high addition of PVM/MA copolymer diminishes this effect.

  • Response: The Vickers hardness test was conducted to determine the effect of adding various amounts of Gantrez towards the surface hardness of PMMA. Statistical analysis was conducted between the various groups and the control group in each time point. The reviewer is asking for clarification as to why the 24 h samples demonstrated more surface hardness than the 30 min samples in each group. Based on our understanding, the error bars of the 30 min and 24 h time points within each group largely overlap, so there is no appreciable difference between the two. If there is a misunderstanding on our part, please let us know.

Suggestion #2: It should be better explained the clear difference in  Wsp/Wsl behavior between the samples with 20% and respectively 25% addition of PVM/MA copolymer, i.e. what explains the sudden increase of solubility. Maybe some evolution curbs vs.  addition of PVM/MA copolymer in all cases would be more effective in understanding the PVM/MA copolymer influence.

  • Response: The second paragraph of 4.3 has been updated based on the reviewer’s suggestions.

Suggestion #3: As regards the antibacterial properties, the preliminary results seems to be promising, but it is obvious that more investigations are needed to conclude on the real impact under real conditions of oral environment.

  • Response: The authors agree with this suggestion of the reviewer, and have clarified in line 347-349 that further antibacterial tests are needed to conclude the impact that real conditions of the oral environment will have on the novel material.

Suggestion #4: On the other hand, the authors should detail the processability of the material with e.g. 10% addition of PVM/MA copolymer comparing to usual polymethyl methacrylate. Even if some general details are provided under 'materials preparation', for current practice such details are considered necessary. 

  • Response: The materials preparation paragraph has been updated based on the reviewer’s suggestions.

Reviewer 3 Report

Dear Authors

In your study, you have investigated a new PMMA formula with Gantrez, an antibacterial copolymer of methyl vinyl ether and maleic acid (PVM/MA). Samples were tested for mechanical properties (surface hardness, flexural strength, water sorption, and water solubility) and effects against Streptococcus mutans. Six groups (0%-control, 5%, 10%, 15%, 20%, and 25% Gantrez) of n = 12 were fabricated for physical property tests. From these results, 3 groups (0%, 5%, and 10% Gantrez) were selected for antibacterial tests. You found that adding 5% and 10% Gantrez into PMMA significantly decreased S. mutans adhesion. There was no significant difference between the control vs. 5%, 10%, 15%, and 20% Gantrez (p > 0.05) for surface hardness, control vs. 5% Gantrez (p > 0.05) for flexural strength, and control vs. 5 and 10% Gantrez for water sorption and water solubility. 
In conclusion, incorporating 5% Gantrez into PMMA presents a promising solution to reduce bacterial adhesion without changing the acrylic resin’s physical properties.

The subject is interesting for the readers and the obtained results are covering an area of health and life quality interest and needs.

However, some comments can be considered for improving the presented work as follows;

General comments

1- The figures' captions are next to the figures

2- The references should be written according to the journal style.

Specific comments

1- 2.2. section: The emersion should be in saliva mimic solution NOT in deionized water.

2- 2.3. section: The equations should be numbered according to the sequence of appearance in the text.

Equation measuring the volume of the samples should identify the  

π value.

3.3. section: line 216 should be corrected and added (5% and 10% Gantrez)

4.1. section: The explanation shoud be supported by experimental measurements such as roughness and SEM examinations.

A major revision is required.

Author Response

Reviewer #3

Dear Reviewer,

Thank you very much for offering your suggestions for our article titled "Antibacterial and physical properties of PVM/MA copolymer-incorporated polymethyl methacrylate as a novel antimicrobial acrylic resin material". We really appreciate your helpful feedback and valuable suggestions. Below is our response to your suggestions:

Suggestion #1: The figures' captions are next to the figures

  • Response: The figures’ captions have been reformatted so that they are next to the figures.

Suggestion #2: The references should be written according to the journal style.

  • Response: After reviewing the journal’s recommended reference style again, we believe the references are formatted according to the journal style.

Specific comments

Suggestion #1- 2.2. section: The emersion should be in saliva mimic solution NOT in deionized water.

  • Response: We understand your concerns regarding using a saliva mimic solution instead of deionized water in the water sorption and solubility tests. During the initial experimental design phase, saliva mimic solution was considered as a possible solution for the various tests conducted. However, the components of saliva mimic solutions are rather variable, and, in the initial test phase of the new Gantrez material, we hoped to keep other possible interfering variables to a minimum. Due on the positive results from this study, the next phase of the experiment will include repeating the tests with artificial saliva.

Suggestion #2- 2.3. section: The equations should be numbered according to the sequence of appearance in the text.

  • Response: The equations have been numbered according to the sequence of appearance in the text.

Suggestion #3: Equation measuring the volume of the samples should identify the  

π value.

·       Response: The π value was estimated as 3.14, and the equation has been updated.

Suggestion #4: 3.3. section: line 216 should be corrected and added (5% and 10% Gantrez)

  • Response: Line 216 has been correct and added.

Suggestion #5: 4.1. section: The explanation should be supported by experimental measurements such as roughness and SEM examinations.

  • Response: In this part of the study the samples were prepared in plastic molds and the materials in the molds were held between between mylar strips and glass slabs before light curing in order to get standard and smoothest surfaces (1,2). Previous studies confirmed that the use of mylar strips provided the smoothest surfaces for resin composite materials. This information was added to the methods part of the manuscript.

  1. Yap A and Mok B. Surface finish of a new hybrid aesthetic restorative material. Oper Dent 2002; 27: 161.
  2. Yap AU, Yap S, Teo C, et al. Finishing/polishing of composite and compomer restoratives: effectiveness of one-step systems. Oper Dent 2004; 29: 275-279

Round 2

Reviewer 1 Report

The author designed and prepared a new PMMA structure, and tested the antibacterial and physical properties of PVM/MA copolymer as a new antibacterial acrylic resin material. The mechanical properties of the sample and its effect on streptococcus showed that there was no significant difference between the sample and the control group, which proved that the author's work was fruitful.

The work has certain innovation and practicality, and also has certain scientific significance. The article is also well organized. This work should have a wide audience, and it is recommended to publish it in this journal.

Reviewer 3 Report

Dear Authors

Thanks for your reply to the comments and for taking them into your consideration during the revision of your manuscript. 

Accordingly, I can recommend your revised manuscript version for publication.